# Somatosensory alpha oscillations gate perceptual learning efficiency

Marion Brickwedde[1,2], Marie C. Krüger[1,2] & Hubert R. Dinse[1,2]

Cognition and perception are closely coupled to alpha power, but whether there is a link between alpha power and perceptual learning efficacy is unknown. Here we show that somatosensory alpha power can be successfully up- and down-regulated with short-term neurofeedback training, which in turn controls subsequent tactile perceptual learning. We find that neurofeedback-induced increases in alpha power lead to enhanced learning, whereas reductions in alpha power impede learning. As a consequence, interindividual learning variability is substantially reduced. No comparable impact is observed for oscillatory power in theta, beta, and lower gamma frequency bands. Our results demonstrate that high pre-learning alpha levels are a requirement for reaching high learning efficiency. These data provide further evidence that alpha oscillations shape the functional architecture of the brain network by gating neural resources and thereby modulating levels of preparedness for upcoming processing.

[1] Neural Plasticity Lab, Institute for Neuroinformatics, Ruhr-University Bochum, 44780 Bochum, Germany. [2] Department of Neurology, BG-University Hospital Bergmannsheil, Ruhr-University Bochum, 44789 Bochum, Germany. Correspondence and requests for materials should be addressed to H.R.D. (email: hubert.dinse@rub.de)

It is a common phenomenon that there are good and poor learners, but the origin of interindividual learning variability remains elusive[1]. Besides obvious attentional and motivational aspects[2,3], quite diverse factors have so far been identified that predict large fractions of learning variability such as genetic polymorphisms[4,5] or cortical gray matter thickness[6]. Recent findings provided evidence that markers of cortical inhibition such as GABA (gamma-aminobutyric acid) levels and somatosensory alpha power (mu rhythm) have a key role in predicting perceptual learning success as well. According to these data, GABA levels explained about 50%[7], and somatosensory alpha oscillations more than 30% of the total learning variance[8]. In addition, it has been reported that pre-training frontal alpha oscillations predicted a considerable amount of subsequent learning in a video game training[9]. EEG alpha oscillations have been subject to extensive research, yet much of its function still remains controversial. The most prominent cortical oscillation measurable with EEG is occipital alpha (8–12 Hz), which occurs over the visual cortex. Additionally, oscillations of this frequency range can be observed over the somatosensory cortex, called somatosensory alpha or mu rhythm. They were discovered in the sixties, where a feline rhythm in the 12 to 20 Hz range has been referred to as sensorimotor rhythm (SMR)[10], which is thought to reflect the human mu rhythm. Initially considered an idling state of the brain[11], today alpha oscillations are assumed to have a critical role in gating information processing by suppressing irrelevant information[12–15]. Numerous studies have shown that optimal alpha power levels positively influence cognitive and perceptual task performance[12,16–22]. However, up to now it remains elusive in how far neuroplasticity and perceptual learning processes benefit from elevated alpha activity as well.

To establish a causal link between individual alpha power and subsequent tactile perceptual learning processes, we took advantage of the fact that alpha oscillations can be purposefully altered by neurofeedback techniques[23,24]. There is agreement that neurofeedback (NF) training can be used to enhance alpha oscillations thereby improving cognitive performance as well as working and episodic memory functions[17–19]. Furthermore, NF training had been successfully applied in the treatment of epileptic seizures and attention deficit hyperactivity disorder[25–27]. However, the modulation of learning processes by NF training has so far not been addressed. We therefore combined NF training to alter somatosensory alpha power (mu rhythm) with a perceptual learning approach. In this way we expected to be able to allow a targeted control of subsequent perceptual learning outcome.

In this work, we demonstrate psychophysically and neurophysiologically the presence of a crucial role for somatosensory alpha power (mu rhythm) relevant for perceptual learning. We find that NF-enhanced alpha power enhances learning efficacy, while NF-reduced alpha power impedes learning. It is argued that alpha oscillations shape the functional architecture of the brain network by gating neural resources and thereby modulating levels of preparedness for upcoming processing.

## Results

**Neurofeedback training of somatosensory alpha power.** We developed a short NF protocol of 30-min sessions on two subsequent days, which enabled participants to alter their somatosensory alpha power. To this aim, we provided color-coded real-time feedback, visualizing the participants' alpha power to enable either an increase (alpha up) or decrease (alpha down) in their oscillatory activity (Fig. 1a). Grand average spectral power changes between first baseline measurements and the last minutes of NF training show significant group differences (two-way mixed ANOVA; main effect time: $F_{(1,45)} = 8.73$; $p < 0.01$; interaction:

$F_{(1,45)} = 8.73$; $p < 0.01$; see Fig. 1c–e; for post-hoc analysis, see Supplementary Data Table 1). While the alpha up group increased their alpha peaks already at the beginning of the first training day, the alpha down group slightly decreased it, staying below their respective baseline level (Fig. 1b). Difficulties to further decrease alpha power levels might be due to floor effects, and the exhausting nature of the alpha down NF training as reported by participants of this group. In contrast, a group of NF-paradoxical-responders (see Methods for details) on average showed no change in alpha peaks. Analysis of the alpha power development in both NF groups for day one of the NF training revealed significant differences over time and between conditions (two-way mixed ANOVA; main effect NF-group: $F_{(1,30)} = 13.46$; $p < 0.001$; main effect NF-block: $F_{(1,30)} = 9.06$; $p < 0.001$; interaction: $F_{(2,60)} = 5.96$; $p < 0.01$; for post-hoc analysis, see Supplementary Data Table 2). On the second day of the NF training, significant differences became apparent between NF groups with slight increases in alpha power levels for both groups (two-way mixed ANOVA; main effect NF-block: $F_{(1,30)} = 5.93$; $p = 0.021$; main effect NF-group: $F_{(1,30)} = 13.84$; $p < 0.001$; for post-hoc analysis, see Supplementary Data Table 3). While we are aware of the limitations arising from using six electrodes only, an analysis of the scalp distribution of alpha power changes in the alpha up group suggests that changes were strongest for the somatosensory cortex (Fig. 1f), while the alpha down group showed an overall decrease of alpha power distributed over the whole scalp (Fig. 1g). As a control for possible unspecific alpha power changes, a third group watched an animal documentary without NF training. Their alpha power levels remained between both NF groups, and increased slightly with no local focus. These results show that our NF protocol enabled participants to successfully alter alpha power levels resulting in substantial group differences within only two days (Fig. 1h).

**Gating of tactile perceptual learning by NF training.** To investigate how altered alpha power levels impact subsequent learning outcomes, we induced a particular form of perceptual learning[28–30] by means of repetitive sensory stimulation[7,8,31–37] (Fig. 2a). This training-independent procedure has been shown to reliably increase tactile acuity of the fingers, unaffected by confounding factors like attention or motivation[34], by mere exposure to high-frequent intermittent finger stimulation[35]. It is a common finding that despite identical input patterns, participants differ substantially in their perceptual learning success[7,8,31–37]. Perceptual changes induced by repetitive sensory stimulation have been shown to be accompanied by major reorganization of the somatosensory cortex, where plastic changes correlate with the amount of improvement in tactile acuity, such as increased BOLD (blood oxygenation level dependent) signals, cortical representational map changes, and gray matter volume[31–33,37].

To assess the magnitude of perceptual learning induced by repetitive sensory stimulation, we implemented a modified version of the two-point discrimination task (2PD; Fig. 2b)[7,8,31–37], where the discrimination threshold does not correspond to the distinction between 1 tip versus 2 tips, but to the distance, when 2 tips are sufficiently separated to be perceived as two (see Methods for details). Figure 2c–h illustrates alpha power levels at baseline and at the end of the NF training as well as baseline psychometric curves and their changes after repetitive sensory stimulation in two representative participants. The first participant shows remarkable training effects in the alpha band (Fig. 2d, e) as well as a distinct perceptual improvement as indicated by a lower discrimination threshold (Fig. 2c; threshold change: 1.65 to 1.3 mm). In contrast, the second participant decreases alpha power levels (Fig. 2g, h), which results in an

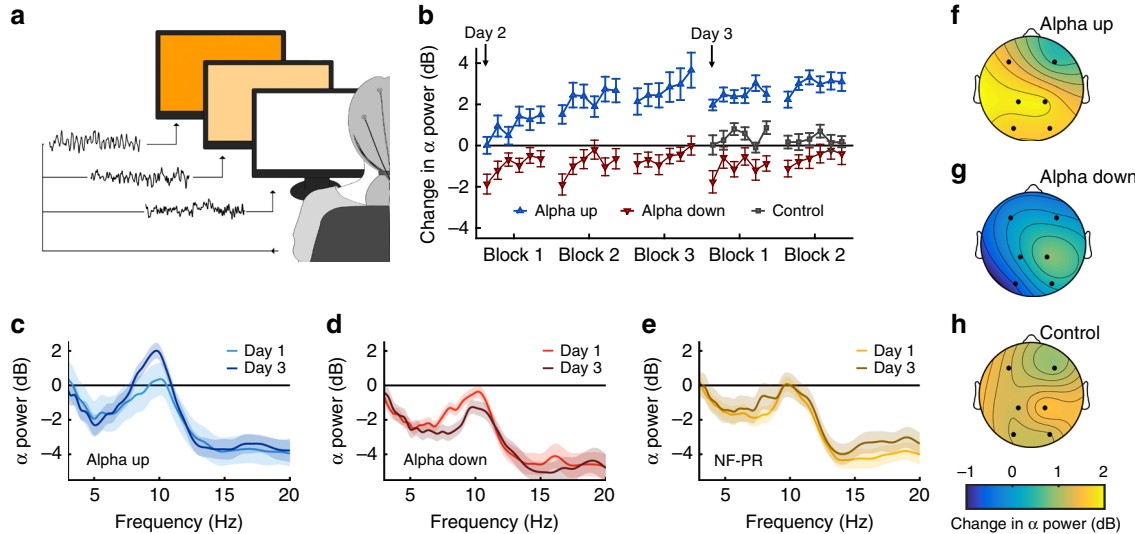

**Fig. 1** Effects of the neurofeedback (NF) protocol on somatosensory alpha power. **a** Illustration of the color-coded real-time visual feedback provided to participants. The screen transiently changes color depending on the amount of alpha power changes over CP1 (deep orange visualized a 10 mV change of alpha power, which could be either an increase or a decrease dependent on the respective NF-group). Shown here is an example for the alpha up group. **b** Time course of alpha power changes on both training days, relative to the first baseline measure on day one (0 dB), revealing considerable training effects. Data are presented as mean ± SEM (standard error of the mean). Each data point represents a 1-min NF training phase. **c–e** Comparison of alpha power spectra between the first baseline on day one and the end of the NF training (power values without baseline normalization) for the alpha up group (**c**) the alpha down group (**d**) and NF-paradoxical-responder (NF-PR; **e**) revealing a clear distinction between groups. Data are presented as mean ± SEM. **f–h** Spectral changes in the alpha peak frequency during NF training on day three, interpolated from the measured electrodes over the scalp for the alpha up (**f**) the alpha down (**g**) and the control (**h**) group (alpha up: $n = 17$; alpha down: $n = 15$; control: $n = 20$; NF-PR: $n = 16$)

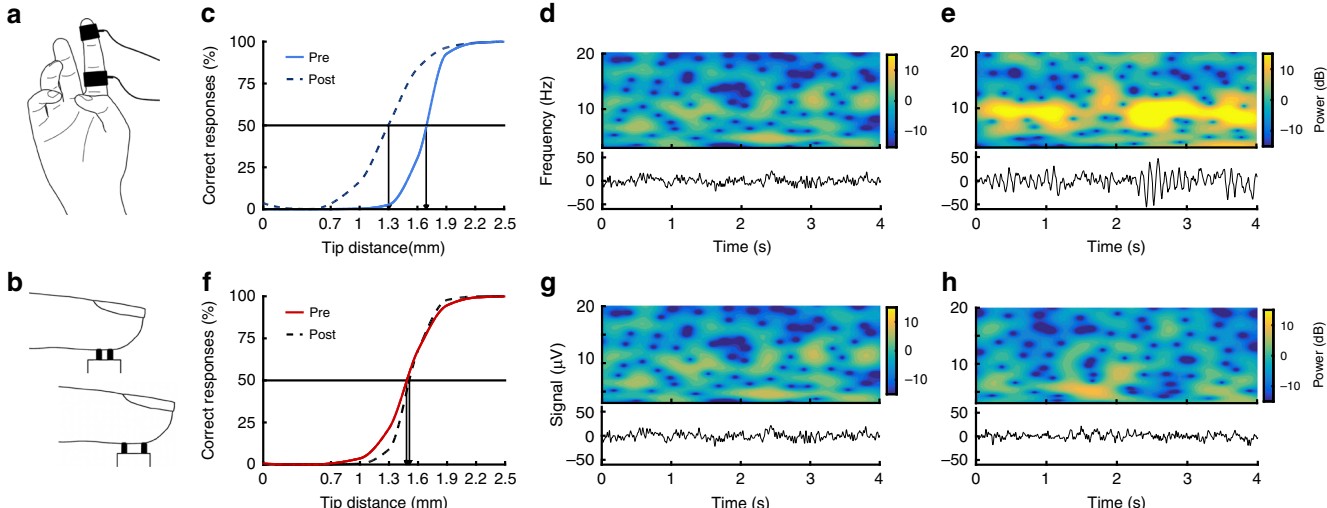

**Fig. 2** Effects of neurofeedback training in two representative participants. **a** Electrical repetitive sensory stimulation on the fingertip elicits perceptual improvement by inducing plastic changes in the somatosensory cortex[31–33, 37]. **b** The two-point discrimination task measures tactile acuity, defined as the minimal distance, where two metal tips presented on the fingertip are correctly perceived as two separate stimuli in at least 50% of the trials. **c** Psychometric curves derived from a participant of the alpha up group. A clear improvement is displayed by the shift of the curve to the left. **d** Time-frequency plots of four seconds of the first measured baseline (top) and oscillatory activity recorded over CP1 (bottom) in the same participant before NF training. **e** At the end of the NF training, a striking increase of power in the alpha band is visible. **f** Psychometric curves derived from a participant of the alpha down group, showing no improvement indicative for impeded learning. **g** Time-frequency plots of four seconds of the first measured baseline (top) and oscillatory activity recorded over CP1 (bottom) in the same participant before NF. **h** At the end of the NF training, decreases in power are present in the alpha band

abolishment of perceptual learning (Fig. 2f; threshold change: 1.49 mm to 1.53 mm).

To quantify these results on a group level, we compiled psychometric curves for each group and applied a two-way mixed *ANOVA* for the thresholds defined as 50% correct responses (Fig. 3a–d). On average, repetitive sensory stimulation induced a discrimination improvement, however, not equally for all groups (main effect pre-post: $F_{(1,64)} = 40.13$; $p < 0.001$; interaction: $F_{(3,64)} = 13.66$; $p < 0.001$). Post-hoc tests (see Supplementary Data Table 4) revealed that participants of the alpha up group, who successfully

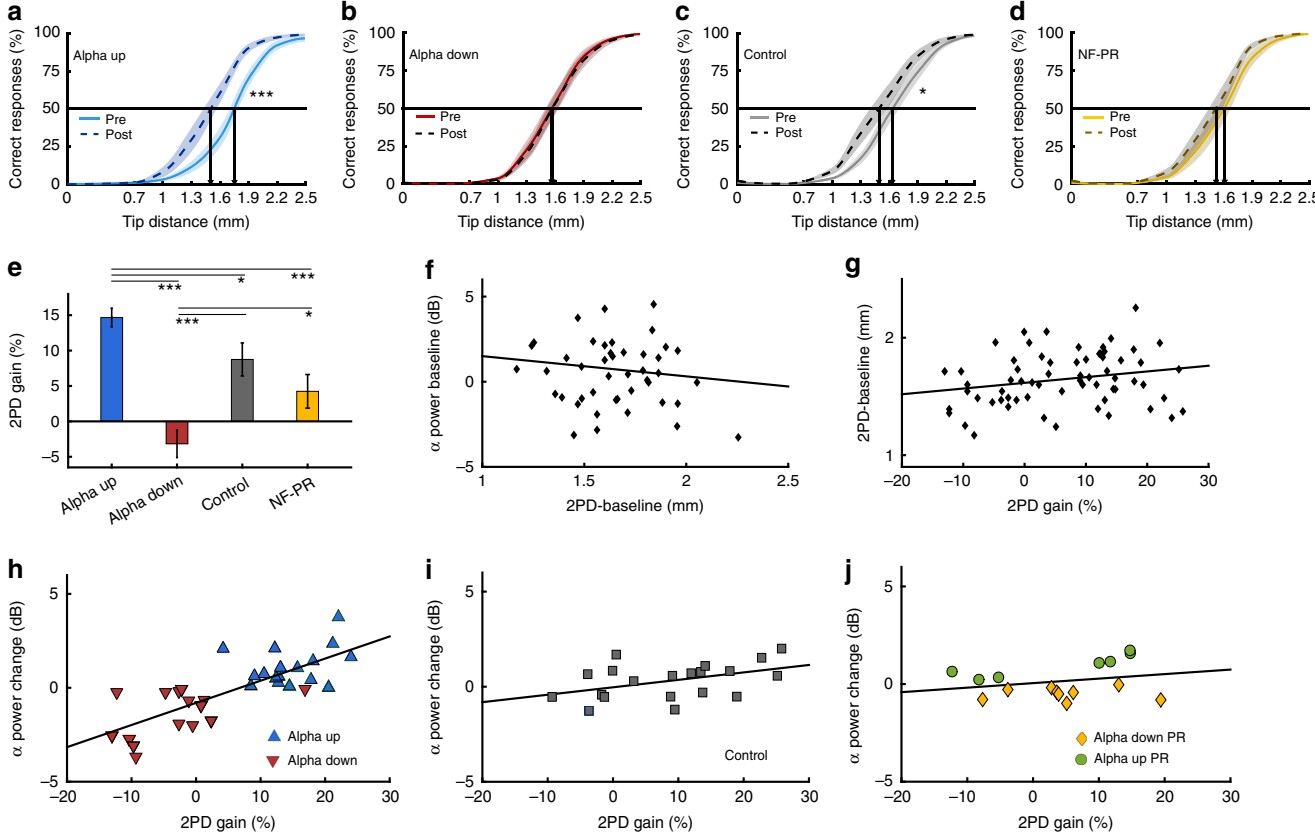

**Fig. 3** Analysis of the effect of neurofeedback training on perceptual learning. **a**–**d** Psychometric curves depicting tactile acuity before and after 20 min of repetitive sensory stimulation. Learning effects are indicated by a shift of the curve to the left. Data are presented as mean ± SEM. *$p < 0.05$; ***$p < 0.001$; two-way mixed ANOVA. **e** Quantitative analysis demonstrating striking group differences in tactile acuity gain. Data are presented as mean ± SEM. *$p < 0.05$; ***$p < 0.001$; one-way ANOVA. **f** No relationship between baseline alpha power and baseline tactile acuity thresholds was detectable. **g** Baseline tactile acuity thresholds and repetitive sensory stimulation-induced tactile acuity gain were equally unrelated. **h** Regression of the percent gain in tactile acuity on alpha power changes compared to baseline for the NF training groups reveals a strong connection. This connection was smaller for participants from the control group (**i**) while NF-paradoxical-responders (NF-PR; **j**) showed no connection at all (alpha up: $n = 17$; alpha down: $n = 15$; control: $n = 20$; NF-PR: $n = 16$). All alpha power values were normalized for baseline values of day 3

increased alpha power, showed the highest tactile acuity changes, whereas participants of the alpha down group, who decreased alpha power, showed no changes in discrimination performance. Participants from the control group exhibited an intermediate improvement usually observed for this kind of repetitive sensory stimulation protocol[7,8,31–37], while NF-paradoxical-responders showed a slight, yet non-significant improvement. Figure 3e displays the extent of perceptual changes compared between groups. Differences between participants from the control group and NF-paradoxical-responders were minor; however, both NF groups differed significantly from them and from each other (one-way ANOVA; $F_{(3,64)} = 12.44$; $p < 0.001$; for post-hoc analysis, see Supplementary Data Table 5). Measurement of alpha levels during the inter-train intervals of the 20 min repetitive sensory stimulation period revealed that NF-induced changes were preserved with no indication for an attenuation of alpha power during the whole duration (two-way mixed ANOVA; main effect group difference: $F_{(1,22)} = 12.836$; $p = 0.002$; for post-hoc analysis, see Supplementary Data Table 6). Accordingly, NF-induced changes of somatosensory alpha power are stable for at least 35 min.

These data demonstrate that we could modulate the perceptual learning outcome induced by repetitive sensory stimulation through NF training in a bidirectional way, with facilitation of learning in parallel to enhanced alpha power but strongly impeded learning in parallel to reduced alpha power. As initially

hypothesized, learning variability as indicated by the standard deviations was strongly reduced (Fig. 3e, h–j), particularly in the alpha up group (alpha up: ±5.4; alpha down: ±7.5). In contrast, standard deviations were higher for the control group (±10.4), the NF-paradoxical-responders (±9.5), and all participants combined (±10.5). Regarding the underlying mechanisms[8,12–15], we suggest that participants of the alpha up group were able to confine instantaneous sensory processing, thereby providing sufficient neural resources to enable more effective subsequent learning. Given that participants of the alpha down group were substantially impeded in their learning, the self-reported fatigue could be compatible with a view of uninhibited information processing, occupying large amounts of neural resources, which in turn become unavailable for subsequent learning processes.

**Effects of electrode locations and other frequency bands.** Independent of global alpha power levels, it has been reported that somatosensory alpha power influences and predicts tactile task performance on a trial by trial basis[20–22]. To rule out that the observed enhancement of tactile performance after repetitive sensory stimulation is simply a consequence of improved processing of sensory afferent inputs due to high alpha power, we analyzed the relationship between baseline tactile acuity and baseline alpha power (Fig. 3f). The lack of correlation provides strong evidence that the improved tactile discrimination is caused

by tactile learning rather than changes of processing efficacy (regression of tactile acuity on baseline alpha power: $p = 0.337$). Furthermore, while not significant, all four groups differed in their baseline tactile acuity. It is possible that baseline performance influences the amount of learning-induced changes. Therefore, we additionally regressed the perceptual learning gain on baseline tactile acuity thresholds (Fig. 3g), and found no discernible connection ($p = 0.104$).

To gain deeper insight into the relation of alpha power and perceptual learning efficiency, we conducted regression analyses between changes of alpha power and changes of discrimination thresholds. There were striking clusters, which distinguish the NF groups (Fig. 3h). These results illustrate the remarkable effect of alpha power, explaining up to 59% of the interindividual variability in perceptual learning outcome (Fig. 3h; $p < 0.001$; $R^2 = 0.59$). For participants from the control group, the same effect is present, although less distinct (Fig. 3i; $p < 0.05$; $R^2 = 0.18$). Surprisingly, NF-paradoxical-responders barely show any perceptual learning, independent of their alpha power (Fig. 3j; $p = 0.343$; $R^2 = -0.002$). It has been shown that periodic alpha fluctuations of oscillatory regimes in the human hippocampus predict a successful performance in working memory maintenance[38]. As the relationship between somatosensory alpha power and perceptual learning varies between our conditions, similar mechanisms could be prevalent in the somatosensory cortex, and critical oscillatory fluctuations might not occur equally in all groups. Rivalling neural processes of higher priority could disrupt fluctuations and induce high somatosensory suppression even during the task, interfering with NF training and perceptual learning. Such processes could, for example, be induced by stress, which has been shown to disrupt perceptual learning[36].

To further analyze the scalp distribution of the explained learning variance, we applied regression analyses between discrimination improvement and alpha power changes recorded at the frontal, occipital, and right-hemispheric electrodes. We found that the variance explained by alpha power is highest over the left somatosensory cortex (59%; Fig. 4a). Explained variance is substantially smaller over the right hemisphere (Fig. 4b; $p < 0.01$; $R^2 = 0.116$) as well as over the left frontal areas (Fig. 4c; $p < 0.001$; $R^2 = 0.171$). Occipital electrodes, however, show no relationship between alpha power and perceptual learning (Fig. 4d; $p = 0.440$;

$R^2 = -0.006$). Our results, together with previous findings[8], give strong indications that alpha power recorded over the somatosensory cortex has a crucial role in tactile learning.

Additionally, we were interested in the influence of other oscillatory frequency bands on perceptual learning and tested the effects of theta, lower and upper beta, as well as lower gamma frequency bands. No substantial learning variance (<10%) could be explained by any of those oscillations (Fig. 4e–h). It is therefore conceivable that alpha power is the most relevant oscillation gating perceptual learning.

## Discussion

Our study demonstrates that high levels of somatosensory alpha oscillations are a requirement for reaching high efficacy in a subsequent perceptual learning task. We developed a neurofeedback protocol to enhance or reduce somatosensory alpha power in human participants within a few sessions over two days. We found that participants in the alpha up group, who successfully increased alpha power, showed the highest gains in a perceptual learning task. In contrast, participants of the alpha down group, who decreased alpha power, showed no changes in performance indicative for a blocking of learning processes. About 59% of the interindividual variability in the perceptual learning outcome was explained by alpha power measured prior to induction of learning. As a consequence, the typically observed high learning variability was substantially reduced. Surprisingly, other oscillatory frequency bands in the theta, lower and upper beta, as well as lower gamma frequency bands explained <10% of the learning variance, implying that alpha power is the most relevant brain oscillation gating perceptual learning.

Research in human subjects demonstrated that training or practicing a task may not be necessary to induce perceptual improvement; however, it can be effectively acquired using a complementary approach in which plasticity processes are driven in response to exposure to repetitive sensory stimulation[30,39]. Repetitive sensory stimulation is an approach that, while applied peripherally, targets the cortical areas that represent the site of sensory stimulation, to facilitate the development of neuroplastic processes[40]. As a result, lasting changes in human perception and in neural processing can be induced without any explicit task training, and without attending to the stimulation. In the

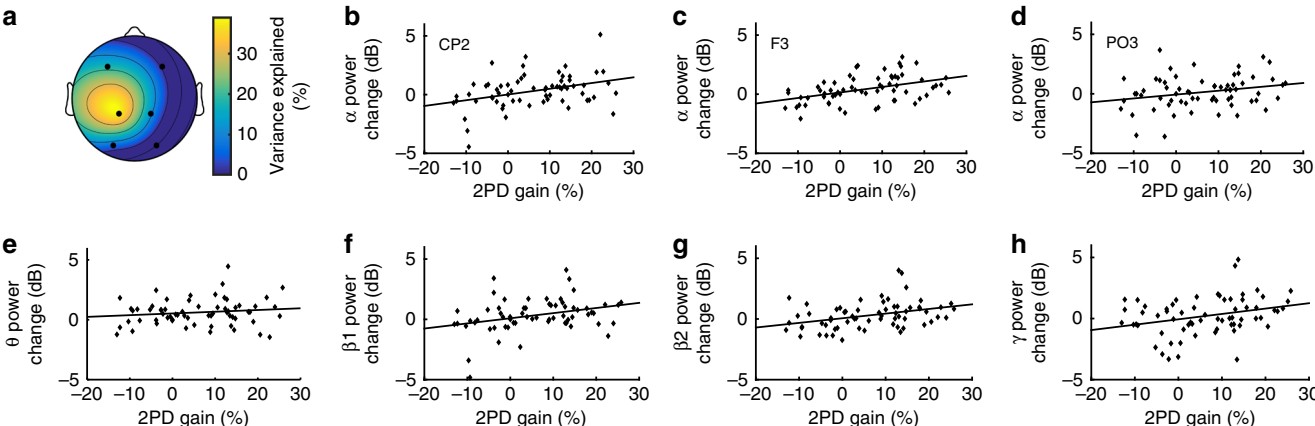

**Fig. 4** Variance explained by electrode position and additional frequency bands. **a** Explained Variance ($R^2$) of the effect of alpha power on perceptual learning interpolated from the measured electrodes over the scalp is highest over the left somatosensory cortex. **b–d** The relationships between perceptual learning and alpha power measured over the right somatosensory cortex (CP2; $n = 66$), over the left frontal hemisphere (F3; $n = 66$), and over the left occipital hemisphere (PO3; $n = 66$) were markedly weaker as compared to the left somatosensory cortex (CP1). **e** Theta oscillations (4–7 Hz) had no measurable effect on perceptual learning ($n = 65$). Minor effects were observable for lower beta (**f** 13–20 Hz; $n = 65$), upper beta (**g**, 21–30 Hz; $n = 67$), and lower gamma (**h** 31–40 Hz; $n = 67$)

somatosensory system, repetitive tactile stimulation has been shown to reliably improve by mere exposure tactile acuity of the fingers, unaffected by confounding factors like attention or motivation[34,41]. To explain this efficacy, this specific form of stimulation was suggested to evoke LTP-like (long-term potentiation) plasticity processes in the cortical regions representing the stimulated skin sites[35,39]. As a result, synaptic transmission is altered and cortical processing is remodeled. The outcome of these processes is reflected in behavioral changes. Evidence from studies in young adult subjects demonstrated major reorganization of the somatosensory cortex including changes in cortical excitability[42], gray matter volume[37], expansion of cortical representational areas[31–33], and enhanced functional connectivity between the somatosensory and motor cortex[43,44]. We here used this approach because learning processes can be induced within a short period of time like 20 min, and because factors related to attention and motivation, which impact learning variability, have no role for this form of learning.

Reducing alpha levels turned out to be more difficult than increasing alpha, which could be ascribed to floor effects. In addition, several subjects from the alpha down group reported fatigue, possibly explaining the poor learning outcome in this group. In case we had used a training-based learning approach, this would have likely been the case. As stated above, the approach of repetitive sensory stimulation has been shown to operate rather independent of attentional and motivational factors[34,41]. These observations favor an explanation, where the lack of efficient learning is related to brain states characterized by low alpha power rather than to mental exhaustion.

To provide an explanation for why within each training block, the alpha down group increased rather than decreased their alpha power, we suggest that keeping alpha low is indeed exhausting. However, after each block, participants had a short break with a brief conversation with the experimenter. After this, there was always another 15-s baseline measure before the next 1-min training started. It is possible that this break was instrumental in enabling the participants to decrease alpha power again.

The subpopulation of NF-paradoxical-responders on average showed no change in alpha peaks, however, closer inspection revealed that the participants from this subpopulation showed a paradoxical behavior, characterized by the fact that alpha up members decreased their alpha, while members of the alpha down group increase it (Fig. 3j). Regarding NF-paradoxial-responders from the alpha down group, we assume that they failed to reduce their alpha power as a consequence of the general difficulty inherent to this task. Instead, when becoming accustomed to the overall experimental situation, which most likely might be accompanied by a general relaxedness, a slight increase in alpha power commonly occurs. A certain degree of relaxedness might also explain the slight increase of alpha levels in some participants from the control group, who watched the animal documentary (Fig. 1b). For NF-paradoxial-responders from the alpha up group, we speculate that these participants were simply unable to increase their alpha power. As a result, they might have become frustrated leading to a stressful situation, which not only interfered with the NF training, but also resulted in a progressive decrease of alpha power over time.

It has been suggested that cortical alpha oscillations gate neural resources and establish a priority system favoring important processes over irrelevant information. In the visual system, research suggested a theory explaining the role of alpha oscillations in information processing[15]. In this framework, alpha oscillations serve an inhibitory function to prevent information overload, as defined by information exceeding the processing capacities of a given system[15]. Alpha oscillations have been shown to rhythmically interrupt high frequency oscillatory bursts in the gamma range (30–150 Hz), which are considered to represent the crucial signatures of ongoing information processing[13,15,45]. According to empirical data and computer simulations, this rhythmic interruption enforces pulsed inhibition which leads to reduced excitability during the oscillatory peaks[13,45–47]. As a result, the time window to process information is limited to the alpha troughs. The order in which different aspects of the stimulus are processed, is then thought to be arranged sequentially, where the phase in the alpha trough in which a gamma burst occurs, carries relevant information. The most salient aspects of a stimulus would be characterized by the high neuronal excitability, firing preferentially at early phases of the alpha cycle, while components with low priority would be cut out by the next alpha peak, not eliciting gamma bursts at all[15].

Given this framework, it is then plausible that the magnitude of alpha power crucially influences the mode of information processing: High alpha power can be expected to reduce the size of the oscillatory troughs, thereby restricting the amount of information that can be processed. On the other hand, low alpha power will increase trough size, thereby providing a dynamic mechanism which controls information processing[13,15,45,47]. In fact, computer simulations of a network model incorporating this framework proved to be successful in predicting neural firing patterns compatible with experimental data[46]. Further studies are required to explore in how far learning processes can be explained within this model.

There are many lines of evidence indicating that top down control impacts alpha power along with task performance[12]. Heightened alpha power levels have for example been observed in disengaged cortical regions during a straining working memory task, thereby focusing information processing on the region most crucial for the task[16]. This task-dependent spatial allocation of neural resources is further complemented by alpha power fluctuations on trial by trial bases, where somatosensory alpha power influences and predicts tactile task performance[22]. For example, low to intermediate pre-stimulus alpha oscillations in the task-engaged cortical areas have been reported to enhance performance in tactile tasks, like temporal discrimination and touch sensitivity[20,21]. Accordingly, pre-stimulus adaptation of alpha levels can be regarded as a preparation for the imminent stimulus, recruiting neuronal resources necessary for efficient processing.

We here suggest that in addition to simultaneous spatial allocation of neuronal resources over cortical areas, and to single trial effects of alpha power on task performance, there is also a global state-effect of alpha power, presumably realized by a state of reduced cortical processing, which preserves neuronal resources for upcoming tasks. Accordingly, heterogeneity in alpha power can be interpreted as variable states of preparedness, leading to high interindividual performance variability in subsequent tasks. Our results show that by rendering alpha power levels through neurofeedback training more homogenous, learning variability can be minimized. This view is in line with numerous studies revealing positive effects of alpha NF training on cognitive performance, by elevating alpha power in advance of cognitive tasks[16–18]. Whether learning and plasticity processes are likewise associated with alpha oscillations so far remained unknown. Recent studies suggested a link between alpha power and perceptual learning. For example, it was reported that pre-stimulus alpha power as well as alpha desynchronization during visual stimulus presentation increased significantly with training[48].

Studies of the feline SMR-rhythm, which is comparable to the human somatosensory alpha (mu) rhythm, identified a thalamocortical loop involving GABAergic inhibition generating the 10 Hz inhibitory cortical oscillations[27]. Cellular studies in macaque monkeys showed that the inhibitory effects of alpha oscillations likely arise from GABAergic inhibition provided by

interneuron and pyramidal cell interaction[49], where interneurons cause fast rhythmic inhibition in the gamma as well as in the alpha range. On the other hand, GABA concentrations in the sensorimotor cortex, assessed by MR spectroscopy, have been reported to predict more than 50% of learning variance[7]. These data are in line with the view that intracortical inhibition and mechanisms preserving a balance of excitation and inhibition seem to be crucially involved in controlling learning processes[50,51]. However, a possible link between markers of inhibition like GABA levels and cortical excitability and alpha oscillations remains so far elusive.

One of the reasons to be interested in alpha oscillations are the prominent role they have in controlling information processing. Our results suggest that in addition high alpha power is implicated in maintaining high perceptual learning efficiency. Accordingly, current theories and findings regarding alpha power are applicable to learning and plasticity processes as well. Most notably, the alpha levels can be controlled easily by training procedures using neurofeedback training, as has already been remarkably demonstrated in clinical applications for treatment of epilepsy and attention deficit hyperactivity disorder[25–27]. This invites further research on the relation between alpha power and other forms of learning beyond perception. Generally, efficient learning is a prerequisite in clinical rehabilitation and in pedagogical education. Accordingly, alpha NF training could be a prime candidate[52] to reduce learning variability and enhance the learning outcome in daily situations[23].

## Methods

**Participants**. In total, 76 healthy volunteers participated in this study (mean age: 24.4 ± 3.1 SD; 36 women). All of them were right-handed as confirmed by the Edinburgh Handedness Inventory[53] (mean laterality quotient: 81.8 ± 19.8 SD), no participants took regular medication (excluding contraceptives). Participants were randomly assigned to two experimental groups and one control group. After completion of the experiment, they received monetary compensation. Eight participants were excluded from further analysis because of insufficient data quality for the following reasons: Participants who fell asleep or closed their eyes for more than two seconds were removed from further analysis ($n = 2$). Furthermore,

participants who were not able to perform the 2PD ($n = 2$) due to poor sensitivity of their fingers, and participants showing strong occipital alpha activity with eyes open, thereby concealing alpha peaks measured at CP1 (according to international 10–20 system; $n = 2$), were also removed. One participant fell ill during experiments and one participant used excessive eye blinking as a strategy during neurofeedback (NF) training. Both were removed from data analysis. Final group sizes consisted of $n = 17$ in the alpha up group, $n = 15$ in the alpha down group, $n = 20$ in the control group, and NF-paradoxical-responders (NF-PR) with $n = 16$ (9 from the alpha up group and 7 form the alpha down group). NF-paradoxical-responders were defined as participants from the alpha up or alpha down group, who on average reduced or increased alpha power, respectively, and were therefore handled as a separate group. The study protocol was approved by the Ethics Committee of the Ruhr-University Bochum and in accordance with the Declaration of Helsinki. All participants provided written informed consent.

**Experimental schedule**. The experiment took place on three consecutive days. On the first day, the tactile acuity task was demonstrated and practiced, and both NF groups underwent a baseline EEG (electroencephalography) measure. On the second day, both NF groups took part in an EEG-baseline measure and subsequent NF training. The last day started with two baseline measures of tactile acuity and one baseline measure of EEG. After this, the NF groups performed a NF training, while the control group watched a muted animal documentary. Immediately afterwards, repetitive sensory stimulation[7,8,31–37] was applied to all participants, while they continued or started to watch the documentary. Finally, tactile spatial acuity was assessed again, serving as a post condition for the effects or repetitive sensory stimulation (Fig. 5).

**Tactile acuity**. Tactile acuity was assessed on the right index fingertip with a modified version of the two-point discrimination task (2PD). It is a two-alternative forced-choice task using the method of constant stimuli[7,8,31–37]. The fingertip was placed on a custom-made device consisting of a rotatable disc with stimuli and an armrest, ensuring standardized assessment. The disc contained 8 stimuli, one with a single tip and seven with two tips separated by varying distances (0.7, 1.0, 1.3, 1.6, 1.9, 2.2, and 2.5 mm). Each stimulus was presented eight times in a pseudorandomized order resulting in a total of 64 trials. Participants reported immediately after the application of the stimulus, whether they perceived one or two stimuli. Opposed to the classical task, where two tips are tested against one, participants had to differentiate between the perception of two clearly separated tips and the perception of two tips still feeling as one for smaller distances. As a marker of tactile acuity, thresholds were defined as the minimal distance with at least 50% correct identifications of two stimuli. Tactile acuity thresholds were estimated by plotting participants' responses against needle distances and fitting them to a psychometric curve using binary logistic regression[7,8,31–37]. It should be noted that our 50% criterion is equivalent to the 75% criterion used in the GOT (grating

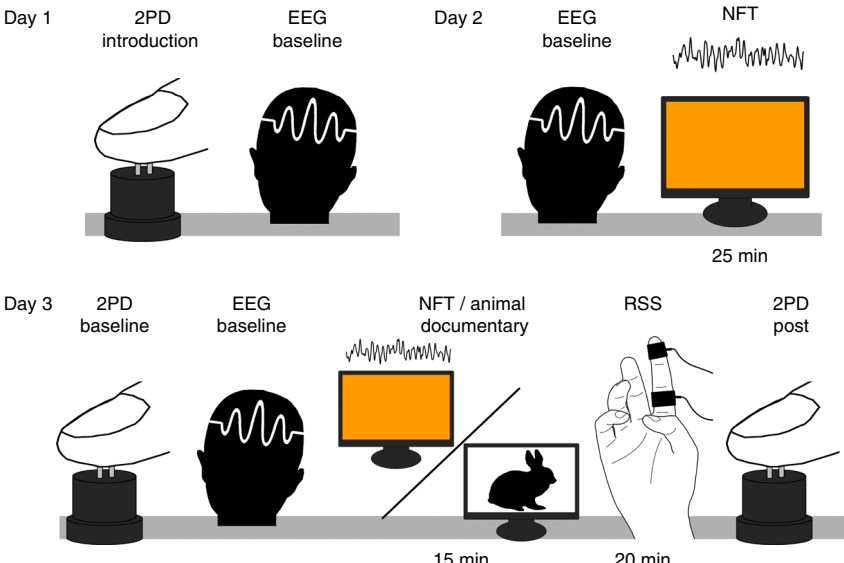

**Fig. 5** Experimental schedule. Display of the experimental schedule over three subsequent days. Day one started with an introductory tactile acuity measure for all groups, followed by an EEG-baseline measure only for both neurofeedback (NF) groups. On day two, both NF groups underwent another EEG-baseline measure and afterwards performed NF training. Day three started with two baseline tactile acuity measures, followed by an EEG-baseline measure for all groups. Afterwards, both NF groups performed NF training, while participants from the control group watched a muted animal documentary. Subsequently, all participants received 20 min of repetitive sensory stimulation (RSS), while beginning or continuing to watch the muted animal documentary. Finally, a last measure of tactile acuity was conducted

orientation task)[54], where 50% is the chance level. The average of both baseline measures on day three was utilized as the 2PD-baseline used for further analyses (test-re-test reliability was high; Cronbach's $\alpha = 0.881$).

**Repetitive sensory stimulation**. Repetitive sensory stimulation was applied to the index finger of the right hand for a duration of 20 min. The stimulation sequence consisted of 20-Hz bursts for 1.4 s with 5 s inter-train intervals, and a ramp/fall time of 0.3 s and 0.2 ms pulse width[35]. The pulse trains were delivered with a stimulation device (ELPHA II 3000, Danmeter A/S). The pulses were transmitted via adhesive surface electrodes fixed to the first and third finger-segment (cathode proximal). The intensity of the stimulation was set individually at the highest threshold values that the participant could easily tolerate for an extended period (range 3–5 mA).

**EEG/neurofeedback**. Both EEG-recordings and NF training were conducted using a 13-channel DC-EEG amplifier (Thera Prax® Mobile, NeuroConn) at a sampling rate of 512 Hz. Participants sat in a comfortable chair inside of a Faraday cage. Before electrode placement, the skin was cleaned with alcohol and prepared with SkinPure preparation gel. The Ag/AgCl electrodes were placed with Elefix conduction gel and arranged according to the international 10–20 system (F3, F4, CP1, CP2, PO3, PO4; ground: forehead; reference: linked mastoids). Additionally, four ocular electrodes were applied. Baseline measures alternated between two eyes-open and eyes-closed conditions each lasting 1 min with a randomized starting condition. The two eyes-open conditions were combined to serve as baseline measure. The combination of the two eyes-closed condition served to identify the occipital alpha peak, as means to differentiate it from the somatosensory alpha peak.

The real-time NF training was adjusted to the alpha peak frequency of the participant, taken from the first baseline measure. Real-time oscillatory power analysis was conducted by applying fast Fourier transformation on sliding 1 s Hann windows with an update rate of 100 ms. The screen visualized the amount of alpha power in this frequency range measured over CP1 compared to baseline with different color saturations from white to orange. White screen color visualized alpha power at baseline levels, while deep orange visualized a 10 mV change of alpha power. This could be either an increase or a decrease dependent on the respective NF-group. One group was trained to increase (alpha up) and the other to decrease (alpha down) alpha power. Both NF groups were blind concerning their condition, and both NF groups were instructed to increase the color saturation of the screen using only their mind.

One block of NF training consisted of a 15 s baseline measurement while fixating a central cross on the screen followed by six training phases. Each of these phases entailed a 1-min training and a 15-s break. On the first NF day, participants trained for three blocks, while on the last day, they trained for two blocks.

**Data processing and analysis**. Ocular artefacts were removed from the EEG signals using least mean squares regression[55]. The corrected signal was manually inspected for remaining artefacts using the EEGLAB toolbox[56]. In total, <5% of the EEG-signal was removed, indicating good data quality. The EEG-signal was filtered between 1 and 40 Hz with a linear finite impulse response (FIR) filter and separated into 2 s epochs. Afterwards, power spectra were extracted using Morlet wavelet convolution (1–25 Hz; 15–25 dynamic cycles) and then averaged over epochs. Baseline normalization was applied with the following formula, where activity marks the EEG-signal of interest and baseline the EEG-baseline used for normalization:

$$10 * \log_{10}\left(\frac{\text{Activity}}{\text{Baseline}}\right) \qquad (1)$$

As the indicator of peak alpha power, the peak of the power spectrum between 8 and 12 Hz was manually chosen.

Time-frequency analysis for representative participants was equally conducted using Morlet wavelet convolution (2–20 Hz; 6–20 dynamic cycles). As baseline data are compared with NF data, no baseline normalization was applied. Instead, to still account for the distortion of $\mu V^2$ power values, the following formula was applied:

$$10 * \log_{10}(\mu V^2) \qquad (2)$$

Signal traces remained unchanged except for filtering and are presented in μV. Mean values, unless specified otherwise, are always reported as mean ± SD.

To assess the stability of alpha power changes, spectral power analysis was performed during inter-train-intervals of repetitive sensory stimulation, dividing the 20 min of stimulation into four sections of 5 min each. The EEG-signal was filtered between 1 and 40 Hz with a linear finite impulse response (FIR) filter and separated into 7 s epochs (the length of one stimulation cycle), with 0 ms marking the onset of stimulation trains. Morlet wavelet convolution (1–25 Hz; 15–25 dynamic cycles) was applied to the window of 2500 to 6500 ms (500 ms before and after the stimulus train) and averaged over epochs. Baseline normalization was applied using formula (1), with the baseline measure recorded on day 1. Participants who were not able to remain still for the duration of the stimulation,

either moving or blinking excessively in more than half of the trials, were removed from this analysis (3 from the alpha up group and 6 from the alpha down group).

Successful implementation of NF training, as well as tactile acuity changes and the stability of alpha power changes, were verified with mixed factorial ANOVAs. In-depth analyses were provided with Bonferroni post-hoc tests. Whether NF training influenced stimulation-induced perceptual learning, was assessed using a one-way ANOVA, applying detailed group analyses with Fisher's LSD post-hoc test. Normal distribution for both types of ANOVAs was confirmed for all entered variables with the Kolmogorov–Smirnov test.

Additionally, power distributions in relation to perceptual learning were inspected with regression analyses. For all electrodes and frequency bands, outliers were excluded, if they diverged more than two SD from the mean of the population. In no case, there were more than two outliers over all participants of all groups combined (outliers—alpha: 0; theta: 2; lower beta: 2; upper beta: 0; low gamma: 0; CP2: 1; PO3: 1; F3: 1). As markers for other frequency bands, maximal values in the specific range were applied (theta: 4–7 Hz, lower beta: 13–20 Hz, upper beta: 21–30 Hz, and lower gamma: 31–40 Hz). Scalp distributions were interpolated using MATLABs griddata function. Regression analyses and ANOVAs were performed in IBM® SPSS® V25; all other analyses were performed in MathWorks® MATLAB R2015a implementing custom code.

## Data availability
The data that support the findings of this study are available from the corresponding author upon reasonable request. A reporting summary for this Article is available as a Supplementary Information file.

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

## Acknowledgements

We acknowledge the support by the Sonderforschungsbereich grant 874 by the German Research Foundation (Deutsche Forschungsgemeinschaft, DFG) to H.R.D., and the International Graduate School of Neuroscience at the Ruhr-University Bochum to M.B. and M.C.K.

## Author contributions

M.B. carried out the experiment, performed the data acquisition and analyzed the data. M.B., M.C.K., and H.R.D. designed the study, interpreted the data, and drafted the manuscript. All authors contributed to the development of the concept of the experiments.

## Additional information

**Competing interests:** H.R.D. hold patents for sensory stimulation procedures. The remaining authors declare that they have no competing interests.

