## [Peer Review File · Nature Communications]

Reviewers' comments:

Reviewer #1 (Remarks to the Author):

Here the authors use neurofeedback training to have subjects either increase or decrease their (global) alpha levels (between subjects design). Then, they test for differences between these groups in a subsequent effect of repetitive sensory stimulation on tactile acuity (this protocol is known to improve acuity, as also shown for control group). The authors find that the "alpha up" group improves acuity (even more than control subjects), whereas this effect is canceled in the "alpha down" group.

I believe this is a very interesting question with potentially promising results, however, the analysis in its current form is poorly executed (or perhaps it is well executed but just poorly reported, it is hard to tell). To me the key point in this story would be to show that neurofeedback significantly altered baseline alpha levels, which then subsequently affects somatosensory perceptual learning (and with topographic specificity!). I don't believe the authors currently show any of that. This is partly due to unclear/confusing description of analysis and results, and partly by missing analyses.

Major comments:

1) It is not clear to me that the changed alpha levels are sustained after the neurofeedback block? If not, what is the proposed mechanism here? The subjects from the respective groups up/down regulate their alpha levels again during the RSS phase? During the acuity test?

2) Perhaps reducing alpha is more difficult and exhausting (the subjects in this group reported fatigue) and this is the full explanation for why they did not learn—they were tired, which might not have anything to do with alpha per se. If you had those subjects do some other demanding and exhausting task, they may not have shown a learning effect either?

3) "Regarding the underlying mechanisms, we suggest that participants of the alpha up group were able to confine instantaneous sensory processing, thereby providing sufficient neural resources to enable more effective subsequent learning."

This is a very specific claim, not at all supported by the data. It remains unclear to me whether the goal of the neurofeedback training was to have subjects specifically target somatosensory alpha, or any alpha whatsoever, and whether the authors were concerned with this distinction at all.

Presumably, if a subject learned to adjust their posterior/visual alpha, rather than their somatosensory/somatomotor alpha, this would have different predicted consequences for a tactile learning task. This appears to not have been taken into account at all. Plotting 6 sensors (?) does not constitute "topographical analysis" (line 93).

4) Please include stats on the neurofeedback training in the Results section. Currently they are only shown in a series of tables in the supplements, without explanation. It would be important to describe the main statistics in the text as well. It is all kind of hard to parse in the current format, and needs a proper description. Since everything that follows hinges on this, I am not quite sure what to make of it.

In any case, I am confused about the stats that were performed. It appears from the tables that baseline levels between groups were only compared on day 1, not on day 2 or 3? Why? And then there is the one random comparison for block 3 day 2? And how come that there is no significant difference between groups (table 1.1), but then when the stats are split up per block (table 2.1) all of a sudden there are significant effects? I also don't understand why this was all split up into different ANOVAs. Tables 2.1/2.2 and table 3 could be more consistent (why is day 2/table 2 split into two

tables but day3/table 3 combined?).

5) Non-responders seems a misnomer as, judging from the description in the Methods section, these subjects did not have a null-effect but rather modulated their alpha in the opposite direction than the group they were ascribed to. Assuming this contained similar numbers of up-group subjects with alpha decrease, and down-group subjects with alpha increase, of course when averaging those together they show no alpha modulation. However, I am not sure that is an accurate depiction. When reading "non-responders" initially I assumed they had no alpha effect in response to neurofeedback training at all.

6) In the Introduction, make it clear from the beginning that this is about somatosensory/tactile learning. It is a bit confusing to read that somatosensory alpha levels predict individual learning abilities/learning processes, when really this is specifically about somatosensory perceptual learning. Currently it reads as if the authors are suggesting a more generalized/generalizable effect.

7) The entire "Data processing and analysis" section could use some work, it is not particularly clear. Non exhaustive lists of comments regarding this section:

- Line 577 "The EEG-signal was filtered between 1 and 40 Hz with a linear finite impulse response (FIR) filter and averaged over 2 s epochs": is this an error or was all analysis performed on an average of the data???? Just to be sure, spectra should be computed on single epochs, then averaged, not the other way around.
- Line 577-578: "As the indicator of peak alpha power, the peak of the spectrum between 8 to 12 Hz was manually chosen". Surely this could have been automated? How about taking the local maximum? Also, this line should come after the description of how spectra were computed.
- I don't understand equation 1. What is "activity" here and what is "baseline"?
- "For other electrodes and frequency bands, outliers were excluded, if they diverged more than two SD from the mean of the population" but not for alpha? Also, where are these data? I did not find them in the Results.
- Were eyes open and eyes closed conditions combined in the analysis? If so, why?

Minor comments:

8) Please specify how many subjects were initially assigned to each NF group.

9) Methods section should also make a clear separation between online analysis for the neurofeedback training (seems to be mostly missing right now) and offline analysis pertaining to the results reported.

10) It would be helpful to the reader to have supplemental figure 1 actually in the text. I had to go back and forth many times to understand the experiment.

11) Please explain in the caption what the different data points are shown in Figure 1b, per block. I am guessing each represents a one minute training phase?

Reviewer #2 (Remarks to the Author):

The paper investigates the effect of central alpha (μ) oscillations neurofeedback on tactile performance. As expected alpha (μ) increase improves tactile implicit skill learning compared to decrease and a control group with visual stimulation. The methodology is clearly described and flawless except that an exact description of the instructions for the different groups is missing: this is

important because the effects reported may simply result from instructional biases, i.e. if subjects used tactile imagination in the up group.

Another weakness is the theoretical framework presented: by ignoring the classical work of Sterman during the sixties, seventies and eighties and collaborators of a thalamic origin of sensorimotor rhythm (SMR) a plausible physiological role of thalamic gating and excitatory-inhibitory cortical balance and GABA interneurons generating the 10Hz inhibitory rhythm at the sensori-motor cortex was provided and the therapeutic effects of SMR neurofeedback training on epilepsy and ADHD were explained. The authors ascribe a preparatory excitation to the alpha (μ) and at the same time discuss the literature of its inhibitory functions: if correct increase in excitation, i.e. measured with slow negativities should correlate with alpha increase, which is not the case.

minor revisions

Reviewer #3 (Remarks to the Author):

The manuscript, "Somatosensory alpha oscillations gate perceptual learning efficiency", by Brickwedde, Krüger, and Dinse, investigates the use of NF training to alter somatosensory alpha power and looking at how that affects tactile perceptual learning. Specifically, the authors hypothesized that they could increase or decrease learning according to whether participants learned to up-regulate or down-regulate their alpha power. And indeed, the authors show that "the participants could modulate the perceptual learning outcome induced by repetitive sensory stimulation through NF training in a bidirectional way, with facilitation of learning in parallel to enhanced alpha power but strongly impeded learning in parallel to reduced alpha power." They also show that this reduces the variation in learning outcomes often seen in PL experiments, and that participants in a control group demonstrate the learning variation that is more commonly seen.

The manuscript was well written and clear and is a compelling follow up to previous studies showing a link between alpha power and PL. Specifically, this study provides useful causal evidence between the two. However, there are a few questions that I would recommend the authors address before the manuscript is published. Assuming these are answered then I would recommend publication. Page numbers below refer to pages in the pdf version of the manuscript.

- p. 3-4, Fig. 1b - Why are the very first trials of the first blocks of the first training day so different between Alpha Up and Alpha Down groups? This should be before any NF training has occurred, correct? Or am I misunderstanding the figure?

- p. 3-4, Fig.1b - The authors comment on the difficulty further reducing alpha levels in the Alpha Down group, but can they comment on why within each training block for the Alpha Down group, their alpha power increases rather than decreases?

- p. 19 Extended Data Tables - It's difficult to interpret what the significant interaction actually means for all of these tables since the ANOVA is a 2x3. A mean plot showing the interaction would be enlightening, although I don't think this is a major issue and is not central to the paper.

- With regard to the overall result: Since the Alpha Down NF training was described as "exhausting", what if exhaustion from trying to down-regulate alpha power caused the lack of learning, not alpha power itself? Explicitly addressing this in the manuscript would be useful.

- An extremely minor concern: I assume the references from the Methods section will be combined with the rest of the references for publication. If not, please combine them because the overlapping numeration between the two sets of references can be confusing.

Reviewers' comments:

Reviewer #1 (Remarks to the Author):

Here the authors use neurofeedback training to have subjects either increase or decrease their (global) alpha levels (between subjects design). Then, they test for differences between these groups in a subsequent effect of repetitive sensory stimulation on tactile acuity (this protocol is known to improve acuity, as also shown for control group). The authors find that the “alpha up” group improves acuity (even more than control subjects), whereas this effect is canceled in the “alpha down” group.

I believe this is a very interesting question with potentially promising results, however, the analysis in its current form is poorly executed (or perhaps it is well executed but just poorly reported, it is hard to tell). To me the key point in this story would be to show that neurofeedback significantly altered baseline alpha levels, which then subsequently affects somatosensory perceptual learning (and with topographic specificity!). I don't believe the authors currently show any of that. This is partly due to unclear/confusing description of analysis and results, and partly by missing analyses.

We thank this reviewer for his critical, but very constructive comments and questions, which turned out being very helpful to improve this manuscript.

Major comments:

1) It is not clear to me that the changed alpha levels are sustained after the neurofeedback block?

If not, what is the proposed mechanism here? The subjects from the respective groups up/down regulate their alpha levels again during the RSS phase? During the acuity test? Fair question. We have a larger study underway where we explore the stability and behavior of alpha during rest (doing nothing), during stimulation and during further testing (such as acuity testing). This is why we did not address this issue. The preliminary data we have so far indicate that NF-induced enhanced or reduced alpha levels are maintained over at least 45 minutes of rest.

To provide information in the context of the present study, and to accommodate the reviewers request, we now show the results of an analysis of alpha power as measured during the interstimulus periods of 5 seconds each when RSS is applied (RSS phase). This analysis confirms that enhanced and reduced alpha levels were preserved over the entire period of 20 minutes. We now present this analysis in Results.

2) Perhaps reducing alpha is more difficult and exhausting (the subjects in this group reported fatigue) and this is the full explanation for why they did not learn—they were tired, which might not have anything to do with alpha per se. If you had those subjects do some other demanding and exhausting task, they may not have shown a learning effect either?

Good point. Indeed, many subjects from the alpha down group reported fatigue, yet not all. And indeed, reducing alpha turned out to be more difficult. Yet, we do not feel that signs of fatigue and exhaustion can easily explain the poor learning outcome. In case we had used training-based learning approaches, this could have likely been the case. But the approach of repetitive sensory stimulation (rss) has been shown to operate rather independent of attentional and motivational factors (cf Godde et al. 2000). Further support for this view comes from an unpublished dataset, where we showed that rss works similarly well when subjects performed an auditory oddball task or a mental calculation tasks thereby distracting

attention. Based on that we favor an explanation, where the lack of efficient learning is related to brain states characterized by low alpha power.

As the latter study is still unpublished, we only added a brief statement in discussion section.

3) “Regarding the underlying mechanisms, we suggest that participants of the alpha up group were able to confine instantaneous sensory processing, thereby providing sufficient neural resources to enable more effective subsequent learning.”

This is a very specific claim, not at all supported by the data. It remains unclear to me whether the goal of the neurofeedback training was to have subjects specifically target somatosensory alpha, or any alpha whatsoever, and whether the authors were concerned with this distinction at all.

Presumably, if a subject learned to adjust their posterior/visual alpha, rather than their somatosensory/somatomotor alpha, this would have different predicted consequences for a tactile learning task. This appears to not have been taken into account at all. Plotting 6 sensors (?) does not constitute “topographical analysis” (line 93).

These are a series of important points, and we are grateful for having the opportunity to clarify these issues.

First, the goal of the neurofeedback training was indeed to have subjects specifically target somatosensory alpha. This was accomplished by using only the alpha power recorded from the left somatosensory cortex CP1 electrode for displaying the alpha level on the feedback monitor. The reviewer is correct that high visual alpha for example would have no effect on tactile learning (but might have an impact on visual learning, which we did not test, of course). Empirical evidence for this is shown in figure 4d, where we show the correlation between tactile perceptual learning and alpha power recorded over left occipital hemisphere (PO3).

As to “topographical analysis”, we readily admit that there are limitations by having used only 6 recording electrodes. So, we replaced “topographical”, or “topography” by “scalp distribution”. Moreover, we added a brief note about the limitations arising from using a 6 electrode setup. However, a true topological analysis was never intended. Instead, what we wanted to show is that selected electrodes over quite different brain regions (frontal, somatosensory, occipital) revealed a very clear regional selectivity of the relation between tactile perceptual learning and alpha power (Fig. 4).

4) Please include stats on the neurofeedback training in the Results section. Currently they are only shown in a series of tables in the supplements, without explanation. It would be important to describe the main statistics in the text as well. It is all kind of hard to parse in the current format, and needs a proper description. Since everything that follows hinges on this, I am not quite sure what to make of it.

Thanks, we agree to this, and moved the F and p values into the text (but left the statistical details in the tables in the supplement). We also included mean plots to all the statistical details, improving the readability.

In any case, I am confused about the stats that were performed. It appears from the tables that baseline levels between groups were only compared on day 1, not on day 2 or 3? Why? And then there is the one random comparison for block 3 day 2? And how come that there is no significant difference between groups (table 1.1), but then when the stats are split up per block (table 2.1) all of a sudden there are significant effects? I also don't understand why this was all split up into different ANOVAs. Tables 2.1/2.2 and table 3 could be more consistent (why is day 2/table 2 split into two tables but day3/table 3 combined?).

We have to admit that this might have created confusion. We were interested in the change of alpha power between the first baseline to the end of neurofeedback training and in possible interactions between groups. This is why we analyzed the first baseline and compared it to the last neurofeedback block. So, when looking at the raw data, we see no difference in the first baseline, however we see an interaction between groups as they change alpha power in different ways.

As for the second analysis, where stats were split up per block, these data were normalized to the baseline on day 1. As such, these data do not depict raw alpha values, but already changes in alpha power. That is why they are suddenly significant, as changes were significantly different between groups.

Your last point is also absolutely valid, the only reason the tables are not consistent, is that one analysis was 2x3 and one was 2x2, which was easier to fit in one table.

We will address the first two points in the text to make it clearer and we will also change the tables to be more consistent.

5) Non-responders seems a misnomer as, judging from the description in the Methods section, these subjects did not have a null-effect but rather modulated their alpha in the opposite direction than the group they were ascribed to. Assuming this contained similar numbers of up-group subjects with alpha decrease, and down-group subjects with alpha increase, of course when averaging those together they show no alpha modulation. However, I am not sure that is an accurate depiction. When reading “non-responders” initially I assumed they had no alpha effect in response to neurofeedback training at all.

Again, thanks for this point. As to the term, we had adopted that from papers about NF training, where they describe “responder” and “non-responder” (i.g. Hanslmayr et al. 2005). To illustrate the true effects of non-responders from both groups, in the scatterplot (Fig. 3j), we now marked the group they were originally assigned to by color. This shows that indeed non-responders did not simply not respond, but did it in the wrong way. We therefore changed the terminology from “non-responders” to “paradoxical responders”, a term used in TMS research. Also, we added some sentences discussing this paradoxical behavior.

6) In the Introduction, make it clear from the beginning that this is about somatosensory/tactile learning. It is a bit confusing to read that somatosensory alpha levels predict individual learning abilities/learning processes, when really this is specifically about somatosensory perceptual learning. Currently it reads as if the authors are suggesting a more generalized/generalizable effect.

Yes, true, thank you. We changed the introduction accordingly.

7) The entire “Data processing and analysis” section could use some work, it is not particularly clear. Non exhaustive lists of comments regarding this section:

- Line 577 “The EEG-signal was filtered between 1 and 40 Hz with a linear finite impulse response (FIR) filter and averaged over 2 s epochs”: is this an error or was all analysis performed on an average of the data???? Just to be sure, spectra should be computed on single epochs, then averaged, not the other way around.

Thanks for pointing this out. In fact, we first computed the spectra of each epoch, and averaged them afterwards. We changed the methods section accordingly, to make that clear.

- Line 577-578: “As the indicator of peak alpha power, the peak of the spectrum between 8 to 12 Hz was manually chosen”. Surely this could have been automated? How about taking the local maximum? Also, this line should come after the description of how spectra were computed.

That is a very important point, warranting explanation. While it is possible to use the local maximum, it for several reasons also introduces additional noise in the data. The somatosensory alpha peak can be distinguished from the occipital alpha peak by comparing eyes-open and eyes-closed conditions. In some participants, we see that even when measuring over the somatosensory cortex, the occipital alpha peak is still higher. Also, peak alpha is not always confined to more or less arbitrary borders of 8 or 12 Hz, as some participants will also have higher or lower frequency alpha peaks. On the other hand, it can happen that theta and beta frequencies (slower and faster oscillations than alpha), fade out into the borders of the alpha range in a spectrum, so that the local maximum would reflect beta or theta oscillations. This is why we chose to look at each participant individually, to correctly identify the somatosensory alpha peak.

- I don't understand equation 1. What is "activity" here and what is "baseline"?
Baseline reflects the EEG-measure, to which you want to normalize the current recording. Activity then reflects the current recording. We now describe this more accurately in the methods section.

- "For other electrodes and frequency bands, outliers were excluded, if they diverged more than two SD from the mean of the population" but not for alpha? Also, where are these data? I did not find them in the Results.

Good point, our description was misleading here. We did not have outliers in all analyses (for example not in the alpha analyses). We now include more detailed information about the outliers in the methods section.

- Were eyes open and eyes closed conditions combined in the analysis? If so, why?
Thank you for pointing this out, we did not describe why we recorded with eyes closed so far. The two eyes-open conditions were combined to constitute the baseline. The two eyes-closed conditions were also combined, to help us identify the occipital alpha peak. We now detail this in the methods section.

Minor comments:

8) Please specify how many subjects were initially assigned to each NF group.
The alpha up group originally included 26 subjects and the alpha down group 22 subjects. We now give these numbers this in the methods section.

9) Methods section should also make a clear separation between online analysis for the neurofeedback training (seems to be mostly missing right now) and offline analysis pertaining to the results reported.
Of course, thank you for pointing this out! We now include descriptions of the online analysis in the methods section.

10) It would be helpful to the reader to have supplemental figure 1 actually in the text. I had to go back and forth many times to understand the experiment.
Yes, we agree to this and moved to figure into the methods section.

11) Please explain in the caption what the different data points are shown in Figure 1b, per block. I am guessing each represents a one minute training phase?
Correct. We changed the caption of Figure 1b to make it clearer.

Reviewer #2 (Remarks to the Author):

The paper investigates the effect of central alpha (μ) oscillations neurofeedback on tactile performance. As expected alpha (μ) increase improves tactile implicit skill learning compared to decrease and a control group with visual stimulation. The methodology is clearly described and flawless except that an exact description of the instructions for the different groups is missing: this is important because the effects reported may simply result from instructional biases, i.e. if subjects used tactile imagination in the up group.

We also thank this reviewer for these positive and constructive comments. As to the first point, we agree that this is an important point. We therefore rewrote this part of the Method Section

Another weakness is the theoretical framework presented: by ignoring the classical work of Sterman during the sixties, seventies and eighties and collaborators of a thalamic origin of sensorimotor rhythm (SMR) a plausible physiological role of thalamic gating and excitatory-inhibitory cortical balance and GABA interneurons generating the 10Hz inhibitory rhythm at the sensori-motor cortex was provided and the therapeutic effects of SMR neurofeedback training on epilepsy and ADHD were explained.

We thank this reviewer for hinting to the seminal work of Sterman. We now cite and discuss his work.

The authors ascribe a preparatory excitation to the alpha (μ) and at the same time discuss the literature of its inhibitory functions: if correct increase in excitation, i.e. measured with slow negativities should correlate with alpha increase, which is not the case.

We are not quite sure what this reviewer means with “preparatory excitation to the alpha”. Is it meant to interpret the increase of alpha as “excitation”? We believe that high alpha leads to a preparatory inhibition, which can be released if an important task begins, leading to high excitation for this task, but inhibiting irrelevant information.

Related to intracortical excitation, we had written that “a possible link between markers of inhibition like GABA levels and cortical excitability and alpha oscillations remains so far elusive”.

Reviewer #3 (Remarks to the Author):

The manuscript, "Somatosensory alpha oscillations gate perceptual learning efficiency", by Brickwedde, Krüger, and Dinse, investigates the use of NF training to alter somatosensory alpha power and looking at how that affects tactile perceptual learning. Specifically, the authors hypothesized that they could increase or decrease learning according to whether participants learned to up-regulate or down-regulate their alpha power. And indeed, the authors show that “the participants could modulate the perceptual learning outcome induced by repetitive sensory stimulation through NF training in a bidirectional way, with facilitation of learning in parallel to enhanced alpha power but strongly impeded learning in parallel to reduced alpha power.” They also show that this reduces the variation in learning outcomes often seen in PL experiments, and that participants in a control group demonstrate the learning variation that is more commonly seen.

The manuscript was well written and clear and is a compelling follow up to previous studies showing a link between alpha power and PL. Specifically, this study provides useful causal evidence between the two. However, there are a few questions that I would recommend the

authors address before the manuscript is published. Assuming these are answered then I would recommend publication. Page numbers below refer to pages in the pdf version of the manuscript.

Many thanks for this very positive evaluation.

- p. 3-4, Fig. 1b - Why are the very first trials of the first blocks of the first training day so different between Alpha Up and Alpha Down groups?

This should be before any NF training has occurred, correct? Or am I misunderstanding the figure?

Thank you for pointing out how this might have been confusing. The very first trials of the first blocks are already the first training session. All data points are normalized to the baseline measured on day 1, so that a value of 0 would be the value of the baseline and values above 0 reflect increases in alpha power, while values below 0 reflect decreases in alpha power. One possible explanation could be that, decreasing alpha power might be easy in the first trial as a new task could be exciting and as such participants would intrinsically already produce less alpha. We will make sure this is clearer in the caption of Figure 1b.

- p. 3-4, Fig.1b - The authors comment on the difficulty further reducing alpha levels in the Alpha Down group, but can they comment on why within each training block for the Alpha Down group, their alpha power increases rather than decreases?

Good point, although we can speculate only. We suggest an explanation based on the assumption that maintaining low alpha is exhausting. After one block, each participant had a short break, where the experimenter talked to the participants, asking if they felt ready to continue and answered final questions. After this, there was always another 15 seconds baseline measure before the next 1 minute training started. It is possible that this intermission helped participants in being able to decrease alpha power again. We now discuss this briefly in the method section.

- p. 19 Extended Data Tables - It's difficult to interpret what the significant interaction actually means for all of these tables since the ANOVA is a 2x3. A mean plot showing the interaction would be enlightening, although I don't think this is a major issue and is not central to the paper.

Good point, thank you. We will include mean plots of interaction in the extended data tables.

- With regard to the overall result: Since the Alpha Down NF training was described as "exhausting", what if exhaustion from trying to down-regulate alpha power caused the lack of learning, not alpha power itself? Explicitly addressing this in the manuscript would be useful. A point also made by reviewer 1. Same answer here:

Indeed, many subjects from the alpha down group reported fatigue, yet not all. And indeed, reducing alpha turned out to be more difficult. Yet, we do not feel that signs of fatigue and exhaustion can easily explain the poor learning outcome. In case we had used training-based learning approaches, this could have likely been the case. But the approach of repetitive sensory stimulation (rss) has been shown to operate rather independent of attentional and motivational factors (cf Godde et al. 2000). Further support for this view comes from an unpublished dataset, where we showed that rss works similarly well when subjects performed an auditory oddball task or a mental calculation tasks thereby distracting attention. Based on that we favor an explanation, where the lack of efficient learning is related to brain states characterized by low alpha power.

As the latter study is still unpublished, we only added a brief statement in discussion section.

- An extremely minor concern: I assume the references from the Methods section will be combined with the rest of the references for publication. If not, please combine them because the overlapping numeration between the two sets of references can be confusing.
Yes, of course, thank you for pointing this out.

REVIEWERS' COMMENTS:

Reviewer #1 (Remarks to the Author):

The authors very adequately and constructively responded to my (substantial) concerns. I believe the manuscript greatly improved with these revisions. I have no further comments.

Reviewer #2 (Remarks to the Author):

The authors changed their manuscript and responded to the criticism of all reviewers, I have no objections anymore to publish it

Reviewer #3 (Remarks to the Author):

I appreciate the clear and well-written replies to my previous comments and the revisions made to the manuscript and its supplemental data. I believe the revisions greatly enhance the manuscript and I have only some minor concerns that I think would improve the manuscript further, all related to the topic of the EEG baseline recording. Once these concerns are addressed then I would recommend the manuscript be published.

- The authors did a good job of clarifying the EEG "baseline" measurement which is being compared against in Figure 1b in their revision, but as a result I'm now a little unclear about Figures 1c-e. I believe the horizontal line representing 0 dB alpha power would be the average alpha power from the first day's baseline recording, correct? If so then the average of the Day 1 line over alpha frequencies should be around that horizontal line, which is indeed the case in Figures 1c and 1e. However, in Figure 1d the Day 1 line is entirely below the 0 dB alpha power line. How is that possible?

- There are multiple baseline recordings, as described in the Methods. Could you clarify which baseline recording is being used for the analyses in Figures 3f-j?

- The "Experimental schedule" section on p. 11 mentions baseline EEG measurements on the first and last day but doesn't mention the baseline recorded on the second day, as illustrated in Figure 5. Could you reconcile those two descriptions of the paradigm?

I'd also like to note that if the editor doesn't feel that these concerns are worth going through another round of revisions and neither of the other reviewers ask for revisions, then I'm happy to allow the current version of the manuscript to be published.

***REVIEWERS' COMMENTS:

Reviewer #1 (Remarks to the Author):

The authors very adequately and constructively responded to my (substantial) concerns. I believe the manuscript greatly improved with these revisions. I have no further comments.

Reviewer #2 (Remarks to the Author):

The authors changed their manuscript and responded to the criticism of all reviewers, I have no objections anymore to publish it

We are very grateful to all reviewers for their positive evaluation.

Reviewer #3 (Remarks to the Author):

I appreciate the clear and well-written replies to my previous comments and the revisions made to the manuscript and its supplemental data. I believe the revisions greatly enhance the manuscript and I have only some minor concerns that I think would improve the manuscript further, all related to the topic of the EEG baseline recording. Once these concerns are addressed then I would recommend the manuscript be published.

We are happy to address the remaining questions about baseline, which indeed had remained somewhat unclear, so thanks for this hint:

- The authors did a good job of clarifying the EEG & baseline measurement which is being compared against in Figure 1b in their revision, but as a result I am now a little unclear about Figures 1c-e. I believe the horizontal line representing 0 dB alpha power would be the average alpha power from the first days baseline recording, correct? If so then the average of the Day 1 line over alpha frequencies should be around that horizontal line, which is indeed the case in Figures 1c and 1e. However, in Figure 1d the Day 1 line is entirely below the 0 dB alpha power line. How is that possible?

Good question. In the case of figures 1c-e, no baseline normalization was applied. This is because we wanted to show the spectrum of the first baseline in comparison to the last block of NF-training right before stimulation. We now clarify this in the figure legend.

- There are multiple baseline recordings, as described in the Methods. Could you clarify which baseline recording is being used for the analyses in Figures 3f-j?

You are right again, we haven't made this clear in the legend of the figure. For the analyses in figures 3f-j, the baseline of day 3 was used, as this provides the best signal-to-noise ratio. This information is now added as well.

- The Experimental schedule section on p. 11 mentions baseline EEG measurements on the first and last day but does not mention the baseline recorded on the second day, as illustrated in Figure 5. Could you reconcile those two descriptions of the paradigm?

Thank you for pointing this out! On the second day, an additional baseline measure was performed, which is now described in the methods section.